# NE contribution to rebooting unconsciousness caused by midazolam

**LeYuan Gu[1,2†], WeiHui Shao[2†], Lu Liu[1†], Qing Xu[1†], YuLing Wang[2†], JiaXuan Gu[2†], Yue Yang[2], ZhuoYue Zhang[1], YaXuan Wu[2], Yue Shen[3], Qian Yu[2], XiTing Lian[1], HaiXiang Ma[4], YuanLi Zhang[2], HongHai Zhang[1,2,3,5]\***

[1]Department of Anesthesiology, Zhejiang University School of Medicine, Hangzhou, China; [2]Department of Anesthesiology, the Fourth Clinical School of Medicine, Zhejiang Chinese Medical University, Hangzhou, China; [3]Department of Anesthesiology, Affiliated Hangzhou First People's Hospital, Westlake University School of Medicine, Hangzhou, China; [4]Medical College of Jining Medical University, Shandong, China; [5]Westlake Laboratory of Life Sciences and Biomedicine, Hangzhou, China

## eLife Assessment

This study provides a **useful** set of experiments showing the relative contribution of the noradrenergic system in reversing the sedation induced by midazolam. The evidence supporting the claims of the authors is **solid**, although specificity issues in the pharmacology and neural-circuit investigations narrow down the strengths of the conclusions. Dealing with these limitations will make the paper attractive to medical biologists working on the neurobiology of anesthesia.

**\*For correspondence:**
zhanghonghai_0902@163.com

[†]These authors contributed equally to this work

**Competing interest:** The authors declare that no competing interests exist.

**Abstract** The advent of midazolam holds profound implications for modern clinical practice. The hypnotic and sedative effects of midazolam afford it broad clinical applicability. However, the specific mechanisms underlying the modulation of altered consciousness by midazolam remain elusive. Herein, using pharmacology, optogenetics, chemogenetics, fiber photometry, and gene knockdown, this in vivo research revealed the role of locus coeruleus (LC)-ventrolateral preoptic nucleus noradrenergic neural circuit in regulating midazolam-induced altered consciousness. This effect was mediated by α1 adrenergic receptors. Moreover, gamma-aminobutyric acid receptor type A (GABAA-R) represents a mechanistically crucial binding site in the LC for midazolam. These findings will provide novel insights into the neural circuit mechanisms underlying the recovery of consciousness after midazolam administration and will help guide the timing of clinical dosing and propose effective intervention targets for timely recovery from midazolam-induced loss of consciousness.

## Introduction

Midazolam has been extensively utilized in clinical practice for nearly half a century. This benzodiazepine exhibits rapid onset and a short duration of action, and causes relatively mild hemodynamic effects (*Mandrioli et al., 2008*; *Reves et al., 1985*). It induces anterograde amnesia, thereby averting intraoperative awareness and mitigating the emergence of malignant memories in patients. The above characteristics of midazolam make it an indispensable drug for the treatment of psychiatric patients in the United States, China, Europe, and numerous other countries worldwide (*Conway et al., 2016*).

Psychiatric disorders globally constitute a significant source of severe, long-term disability and socioeconomic strain. Among these, insomnia stands as a prevalent health issue in the general population and clinical settings (*Perlis et al., 2022*). The World Health Organization reported that about

30% of the adult population worldwide suffer from insomnia, especially alongside other mental and physical health conditions (*Sutton, 2021*). Intractable insomnia frequently co-occurs with severe complications, including cognitive and immune decline, emotional disorders, and suicidal tendencies (*Buysse, 2013*; *Gebara et al., 2018*). Oral benzodiazepines, exemplified by midazolam, constitute the foundation of their therapy regimen (*Riemann et al., 2015*). Furthermore, agitation, a prevalent clinical issue in numerous psychiatric disorders, significantly compounds morbidity during hospitalization (*Citrome, 2021*). The extensively used medication for patients with acute agitation in the emergency department and intensive care unit is midazolam (*Kim et al., 2021*; *Shafer, 1998*). In conclusion, midazolam has emerged as one of the most commonly administered psychotropic drugs in clinical practice, effectively inducing a sedative response with diminished consciousness.

However, a critical challenge in the clinical application of midazolam pertains to its safety profile. Depending on the dose, it produces effects ranging from anxiolysis to loss of consciousness, leading it to be versatile to have anxiolytic, sedative, and hypnotic effects (*Kanto, 1985*; *Olkkola and Ahonen, 2008*). Prolonged or excessive midazolam administration frequently results in increased sedation accumulation and depth, potentially causing delayed recovery, which in turn prolongs hospitalization and precipitates diverse complications (*Shehabi et al., 2012*). Consequently, it is imperative to investigate the mechanisms underlying midazolam-induced altered consciousness and find promising targets to prevent its complications.

Midazolam has depressant effects on the central nervous system (CNS). It is believed to act on the brainstem reticular formation and limbic system via gamma-aminobutyric acid receptor type A (GABAA-R), reducing brain excitability and inducing sedation. GABAA-R is the initiating molecular target of midazolam action. However, the specific mechanism governing the regulatory systems downstream of the GABAergic system in midazolam-induced alterations in consciousness remains to be elucidated. Midazolam elicits an electroencephalographic (EEG) pattern resembling normal non-rapid eye movement (NREM) sleep, lacking a sustained long-range-specific activation pattern compared to wakefulness (*Massimini et al., 2012*). Therefore, we postulate that a midazolam-induced loss of consciousness, at least partially, involves the activation of an endogenous sleep-promoting pathway.

The locus coeruleus (LC) serves as the primary site for norepinephrine (NE) synthesis and release in the brain, exhibiting extensive projections to various other brain regions (*Benarroch, 2018*). The LC[NE] neurons, which have been linked to multiple functions, including sleep and arousal, stress-related behaviors, attention, and pain conduction, are reportedly instrumental in sleep–arousal regulation (*Poe et al., 2020*; *Suárez-Pereira et al., 2022*). Optogenetic activation of LC[NE] neurons results in an instantaneous shift from sleep to wakefulness (*Carter et al., 2010*). The ventrolateral preoptic nucleus (VLPO) in the preoptic hypothalamus is recognized as a pivotal 'sleep center' in which sleep-activated neurons orchestrate the initiation and promotion of sleep, and these neurons are subject to regulation by the LC[NE] neurons (*Chou et al., 2002*; *Sherin et al., 1996*; *Sherin et al., 1998*). Recently, it has been reported that optogenetic activation of the LC-VLPO NEergic neural circuit facilitates arousal from sleep, mediating its effect through distinct adrenergic receptors, indicating the significance of this neural circuit in controlling wakefulness (*Liang et al., 2021*). However, natural sleep is a physiological state, whereas drug-induced loss of consciousness is a pathological state, and the two are quite different from a pathophysiological perspective despite some correlation. Furthermore, dexmedetomidine-induced sedation encompasses the utilization of the endogenous sleep-promoting pathway mediated by the LC-VLPO (*Nelson et al., 2003*). Therefore, these studies offer hints for additional investigation into the role of LC-VLPO NEergic neural circuit in midazolam-induced alterations in consciousness. This will facilitate a more targeted clarification of the neural mechanisms underlying the sedative-hypnotic effects of midazolam, with profound clinical implications.

In the current study, it was unveiled that midazolam initially functioned by acting on GABAA-R in the LC. Crucially, we found that the LC-VLPO NEergic neural circuit plays an important role in promoting recovery from midazolam and that this effect was primarily mediated by α1 adrenergic receptors (α1-R). We took a cascading approach from peripheral to central to validate our hypothesis. Initially, we used pharmacological methods to investigate the link between the NE system and awakening from midazolam administration. Subsequently, we adopted more precise and targeted approaches, such as optogenetics and chemogenetics, to further substantiate our finding and successfully prove the involvement of the NEergic LC-VLPO neural circuit in facilitating the restoration of consciousness following midazolam administration. In addition, we employed both female and male mice in a series

of chemogenetic activation experiments, suggesting that interventions that promoted rebooting from midazolam-induced unconsciousness in male mice were equally effective in female mice. In other words, our findings possess universal applicability and hold significant clinical application value. Based on the above experimental results, we further employed the knockdown technique to specifically manipulate GABAA-R on LC^NE neurons. We found that knockdown GABAA-R not only reduced recovery time from midazolam but also affected calcium signals in NEergic terminals at VLPO. More importantly, specifically block α1-R at VLPO reversed the effect of shortened recovery time produced by GABAA-R knockdown on LC^NE neurons.

These results will significantly enhance our understanding of the neural circuit mechanisms of midazolam-induced altered consciousness and provide a potential target for interventions aimed at addressing delayed recovery caused by midazolam.

## Results

### The noradrenergic system contributes to recovering from midazolam administration

Before starting the following experiments, we first determined the optimal intraperitoneal (IP) dose of midazolam-induced sedation (loss of consciousness). We chose the lowest effective dose that could induce loss of righting reflex (LORR) successfully in all of the mice, that is, 60 mg/kg, as the dose for the subsequent experiments (*Figure 1A and B*).

To examine the role of the central noradrenergic system, we individually analyzed the NE content in the prosencephalon and brainstem of mice (*Figure 1C*). Given that NE in the brain is mainly derived from the brainstem, the NE content in the prosencephalon differed significantly from that in the brainstem under normal conditions (p<0.0001, *Figure 1D*). However, we also found that compared with the control group, the NE content in the brainstem was significantly reduced after midazolam administration (p<0.05, *Figure 1D*).

Subsequently, we adopted a pharmacological method of peripheral intervention on NE. We found that IP injection of atomoxetine (20 mg/kg), a central selective NE reuptake inhibitor, increased the number of c-Fos (+)/TH (+) cells in the LC (p<0.01, *Figure 1J and L*) and shortened the recovery time from midazolam administration (p<0.05, *Figure 1F*). In contrast, IP injection of N-(2-chloroethyl)-N-ethyl-2-bromoben-zylamine hydrochloride (DSP-4) (50 mg/kg), a highly selective NEergic neurotoxin, reduced the number of TH+ cells in the LC, most significantly on day 10 after injection (p<0.0001, *Figure 1I and K*) and prolonged the recovery time from midazolam administration (p<0.05, *Figure 1G*). Furthermore, continuous IP injection of DSP-4 for 10 days significantly reversed the recovery-promoting effects of atomoxetine (p<0.01, *Figure 1H*). These findings collectively indicate that the noradrenergic system contributes to the midazolam-induced altered consciousness.

### The activity of LC^NE neurons is significantly reduced after midazolam administration

Since LC is the largest nucleus in the brain that syntheses and releases NE, we utilized fiber photometry to quantify the neuronal activity of LC^NE neurons at different states after midazolam administration (*Figure 2A and B*). We injected AAV2/9-Dbh-GCaMP6m-WPRE-hGH pA with a specific promoter into the bilateral LC of C57BL/6J mice to express the GCaMP6m calcium indicator specifically in NEergic neurons and examined changes in GCaMP6m fluorescence signal through an optical fiber located in the LC (*Figure 2C and D*). During the LORR to recovery of righting reflex (RORR) phase, we found that the ΔF/F peak of calcium signaling in LC^NE neurons was significantly reduced compared with other phases, and then increased after RORR (p<0.05, p<0.01, *Figure 2E–H*).

To further evaluate the changes in neuronal activity in the LC following midazolam administration, we performed immunofluorescence staining for c-Fos and TH. Fluorescence images revealed a reduction in the number of c-Fos-positive cells co-labeled with TH in the midazolam administration group compared to the control group (p<0.01, *Figure 2I–K*). These results indicated that LC^NE neurons were significantly inhibited after midazolam administration.



**Figure 1.** The noradrenergic system is involved in recovery from midazolam administration. (**A**) Number of LORR and no LORR induced by different doses of midazolam in C57BL/6J mice. (**B**) Rate of LORR (%) in C57BL/6J mice at different doses of midazolam. (**C**) Protocol for investigating changes in the content of NE in the prosencephalon and brainstem of C57BL/6J mice by ELISA. (**D**) Content of NE in the prosencephalon and brainstem in the

*Figure 1 continued on next page*

*Figure 1 continued*

vehicle and midazolam (60 mg/kg, i.p.) groups. (**E**) Protocol for exploring the influence of i.p. injection of DSP-4 on the atomoxetine-mediated shortening of the recovery time from midazolam administration. (**F**) Comparison of recovery time in the vehicle, atomoxetine (10 mg/kg, i.p.), and atomoxetine (20 mg/kg, i.p.) groups. (**G**) Comparison of recovery time in the vehicle, DSP-4 (50 mg/kg, 3 days before, i.p.), and DSP-4 (50 mg/kg, 10 days before, i.p.) groups. (**H**) Comparison of recovery time in the vehicle + vehicle, vehicle + atomoxetine, DSP-4 (3 days before) + atomoxetine, and DSP-4 (10 days before) + atomoxetine groups. (**I**) Comparison of normalized TH (+) cell number in the LC in the vehicle, DSP-4 (3 days before), and DSP-4 (10 days before). (**J**) The quantification of c-Fos (+)/TH (+) cells with or without i.p. injection of atomoxetine. (**K**) Images of TH+ neurons in the LC after i.p. injection of vehicle, DSP-4 (3 days), or DSP-4 (10 days) (panels on the lower image show magnified images of the panels on the upper image). (**L**) Representative images showing the changes in TH (+) neuronal activity with or without i.p. injection of atomoxetine. Mida: midazolam; NE: norepinephrine; i.p.: intraperitoneal injection; LORR: loss of righting reflex; RORR: recovery of righting reflex; LC: locus coeruleus; TH: tyrosine hydroxylase; *p<0.05; **p<0.01; ***p<0.001; ****p<0.0001.

## LC^NE neurons contribute to regulating recovery from midazolam

To delve deeper into LC^NE neurons in recovery after midazolam administration, we artificially intervened in LC^NE neurons. We microinjected DSP-4 into LC to specifically degrade LC^NE neurons and found a significant decrease in the number of TH+ neurons in the LC of mice 10 days after the DSP-4 microinjection (p<0.0001, *Figure 2P and Q*). Furthermore, microinjecting DSP-4 into LC not only significantly extended the recovery time from midazolam-induced loss of consciousness but also attenuated the recovery-promoting effect of intraperitoneally administered atomoxetine (p<0.05, *Figure 2N and O*). These suggest that intranuclear administration of DSP-4 to specifically ablate LC^NE neurons postpones recovery from midazolam administration.

Then, optogenetic manipulation was used to further examine the role of LC^NE neurons (*Figure 3A and B*). For this purpose, we combined a Cre-dependent AAV expressing channelrhodopsin-2 (ChR2) fused with enhanced yellow fluorescent protein (AAV-DIO-ChR2–eYFP) with Cre-AAV expressing the TH promoter (Th-Cre-AAV) to restrict the expression of ChR2 to LC TH+ neurons and an optrode was implanted at the injection site (*Figure 3C and D*). Our results showed photostimulation (20 min/4 mW) of LC^NE neurons significantly shortened the recovery time and produced a pro-recovery effect from midazolam regardless of whether it was left, right, or bilateral LC optogenetic activation (p<0.05, *Figure 3E–G*, *Videos 1 and 2*). Additionally, fluorescence images revealed a significant increase in the number of c-Fos (+)/TH (+) cells after photostimulation of LC^NE neurons (p<0.0001, *Figure 3H–J*).

Next, we tested whether the chemogenetic activation of LC^NE neurons promotes recovery (*Figure 3K*). In order to reveal whether there was sex discrepancy in chemogenetic activation of LC^NE neurons, we used both male and female mice, and Cre-dependent excitatory DREADD was co-infused with Th-Cre-AAV to restrict the expression of hM3Dq to LC TH+ neurons (*Figure 3L*). We determined the transfection efficiency of the chemogenetic virus in LC^NE neurons (*Figure 3M–O*). Similar results were obtained in the chemogenetic experiment. We found that in both male and female mice, IP injection of 0.2 mg/kg CNO could shorten the recovery time and produce a pro-recovery effect from midazolam regardless of whether it was left, right, or bilateral LC chemogenetic activation (p<0.05, *Figure 3P–S*, *Videos 3–6*). Additionally, c-Fos (+)/TH (+) cells were significantly increased after photostimulation of LC^NE neurons (p<0.0001, *Figure 3T–V*). Thus far, these findings have proven that there was no sex discrepancy in chemogenetic activation of LC^NE neurons, and LC^NE neurons contribute to the recovery from midazolam.

## Activation of the LC-VLPO NEergic neural circuit promotes recovery from midazolam

The LC sends projections to several brain regions, including the VLPO, which is a crucial area in regulating sleep and arousal states. Therefore, we hypothesized that LC and VLPO interact to mediate midazolam-induced alterations in consciousness. To confirm the existence of the LC-VLPO projection, we microinjected the mix of mTh-Cre-AAV and AAV-EF1a-DIO-hChR2(H134R)-eYFP viruses into the LC and microinjected rAAV2/9-DβH-GCaMP6m-WPRE-hGH pA into the VLPO. Two weeks later, optical fibers were implanted into the VLPO to record neuronal firing (*Figure 4A and B*). We observed local NEergic terminals within the VLPO (*Figure 4D and K*). Also, we found that optogenetic or

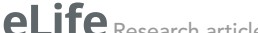

**Figure 2.** LC^NE neuronal activity is significantly reduced after midazolam administration and selective ablation of LCNE neurons hinders recovery from midazolam. (**A**) A schematic of the fiber optic recording of calcium signal in the LC. (**B**) A schematic of the calcium signaling recording device. (**C**) A representative photomicrograph showing the microinjection and optical fiber locations and the co-expression of GCaMP6s and TH. (**D**) Quantification of

*Figure 2 continued on next page*

*Figure 2 continued*

the percentage of GCaMP6m (+)/TH (+) colocalization in all TH (+) neurons in LC. (**E**) Schematic diagram of the method of dividing the entire recording of the calcium signal at the LC into four stages, and sources of heatmap and ΔF/F(%) statistical data. (The time points for awake, midazolam injection, LORR, and RORR in mice were respectively selected as time = 0. The selected traces lasting for 20 s were based on the length of a complete Ca$^{2+}$ signal.) (**F**) A heatmap of calcium signaling changes in bilateral LC$^{NE}$ neurons induced by midazolam. (Two representative mice were selected for the fiber optic recording of calcium signals, and the whole recording was divided into four stages, each stage as a group for statistical analysis of calcium signaling.) (**G**) A statistical diagram of calcium signaling changes in bilateral LC$^{NE}$ neurons induced by midazolam. (**H**) The peak ΔF/F in the wakefulness, before LORR, LORR-RORR, and after RORR stages. (**I, J**) Representative images showing the co-expression of c-Fos and TH in the LC without or with midazolam treatment. (**K**) The quantification of c-Fos (+)/TH (+) cells in the LC with midazolam or without midazolam treatment. (**L**) Protocol for exploring the influence of intra-LC microinjection DSP-4 on the atomoxetine-mediated shortening of the recovery time from midazolam administration. (**M**) The representative photomicrograph shows the tracks of cannulas implanted into bilateral LC. (**N**) Results show the effects of microinjection of DSP-4 (10 days before) into bilateral LC on the recovery time of midazolam. (**O**) Results show the effects of vehicle + vehicle, vehicle + atomoxetine, and DSP-4 (10 days before) + atomoxetine on the recovery time of midazolam. (**P**) Comparison of normalized TH (+) cell number in the LC with or without DSP-4 (10 days before) microinjected in the bilateral LC. (**Q**) Images of TH+ neurons in the LC after microinjection of DSP-4 or vehicle for 10 days (panels on the right show magnified images of the panels on the left). LC: locus coeruleus; i.p.: intraperitoneal injection; LORR: loss of righting reflex; RORR: recovery of righting reflex; TH: tyrosine hydroxylase; *p<0.05; **p<0.01; ****p<0.0001.

chemogenetic activation of LC$^{NE}$ neurons resulted in an enhanced ΔF/F peak in the VLPO (p<0.05, p<0.01, *Figure 4O–X*), that is, increased VLPO activity, as well as facilitated recovery from midazolam (p<0.05, *Figure 4G and M–N*). Moreover, we found that there was no sex discrepancy in chemogenetic studies.

To further identify whether the LC-VLPO NEergic circuit mediates promoting recovery, we used optogenetics to activate LC$^{NE}$ terminals in the VLPO. We injected the mixture of Dbh-Cre-AAV and AAV-EF1a-DIO-hChR2(H134R)-eYFP viruses into bilateral LC and AAV-EF1a-DIO-hChR2(H134R)-eYFP virus into bilateral VLPO, then implanted optical fibers above the VLPO (*Figure 5A and B*). Immunofluorescence results showed expression of hChR2 in the VLPO (*Figure 5F*). It confirms the structural association between the LC and the VLPO. In addition, when the NEergic terminals of the LC-VLPO pathway were photostimulated with blue light, we observed a shorter recovery time in the VLPO photostimulation group compared to the non-photostimulation group (p<0.01, *Figure 5G*). Thus, optogenetic activation of VLPO-projecting LC$^{NE}$ neurons promotes recovery from midazolam. Next, we tested whether the chemogenetic activation of LC$^{NE}$ terminals in the VLPO promotes recovery (*Figure 5H and I*). We injected the mixture of Dbh-Cre-AAV and AAV-Ef1a-DIO-hM3Dq-mCherry viruses into bilateral LC and AAV-Ef1a-DIO-hM3Dq-mCherry virus into bilateral VLPO, then implanted cannula above the VLPO to inject CNO (20 ug/mL, 400 nL). In order to reveal whether there was sex discrepancy in chemogenetic activation of LC$^{NE}$ neurons, we used both male and female mice (*Figure 5J–N*). Similar results were obtained in the chemogenetic experiment. We found that in both male and female mice, intra-VLPO microinjection of CNO could produce a pro-recovery effect from midazolam (p<0.05, *Figure 5N*). Thus far, our results show the contribution of the LC-VLPO NEergic neural circuit in regulating midazolam-induced altered consciousness.

## The promotion of recovery from midazolam is primarily mediated by the α1 adrenergic receptor

To identify the role of different types of noradrenergic receptors in the brain, we used a pharmacological approach. By intracerebroventricular (ICV) injection of different doses of α1-R, α2-R, and β-R agonists and antagonists (*Figure 6A*), we found that the α1 receptor agonist phenylephrine (20 mg/mL) significantly shortened recovery time from midazolam (p<0.05, *Figure 6C*). Conversely, the α1 receptor antagonist prazosin (1.5 mg/mL), the α2 receptor agonist clonidine (1.5 mg/mL), and the β receptor antagonist propranolol (5 mg/mL) caused the opposite effects (p<0.05, *Figure 6D, E and H*). However, the α2 receptor antagonist yohimbine and the β agonist isoprenaline did not affect the recovery time (p>0.05, *Figure 6F and G*), and phenylephrine significantly reversed the effect of



**Figure 3.** Optogenetic or chemogenetic activation of LC^NE neurons promotes recovery from midazolam. (**A**) Protocol for the effect of PS of LC^NE neurons with different light parameters on the recovery time of midazolam. (**B**) A schematic of the optogenetic instrument. (**C**) A representative photomicrograph showing the locations of optical fiber and virus expression and the co-expression of hChR2 and TH. (**D**) Quantification of the percentage

*Figure 3 continued on next page*

*Figure 3 continued*

of eYFP (+)/TH (+) colocalization in all TH (+) neurons in LC. (**E, F**) Results show the effect of PS of LC$^{NE}$ neurons with different light parameters on the recovery time of midazolam. (**G**) Results show the effect of left, right, and bilateral PS of the LC$^{NE}$ neurons on the recovery time of midazolam. (**H**) The quantification of c-Fos (+)/TH (+) cells in the LC with or without PS. (**I, J**) Representative images showing the co-expression of c-Fos and TH cells in the LC with or without PS. (**K, L**) Protocol and instrument for the effect of chemogenetic activation of LC$^{NE}$ neurons on the recovery time of midazolam. (**M**) A representative photomicrograph showing the virus expression and the co-expression of mCherry and TH in the LC of male and female mice. (**N, O**) Quantification of the percentage of mCherry (+)/TH (+) colocalization in all TH (+) neurons in LC. (**P, Q**) Results show the effect of chemogenetic activation of LC$^{NE}$ neurons with different doses of CNO (i.p.) on the recovery time of midazolam in male and female mice. (**R, S**) Results show the effect of left, right, and bilateral chemogenetic activation of the LC$^{NE}$ neurons on the recovery time of midazolam in male and female mice. (**T**) Representative images showing the co-expression of c-Fos and TH cells in the LC of male and female mice with or without intraperitoneal injection of CNO (0.2 mg/ kg). (**U, V**) The quantification of c-Fos (+)/TH (+) cells in the LC with or without CS. LC: locus coeruleus; i.p.: intraperitoneal injection; LORR: loss of righting reflex; RORR: recovery of righting reflex; TH: tyrosine hydroxylase; PS: photostimulation; CS: chemogenetic stimulation; *p<0.05; ****p<0.0001.

propranolol in prolonging recovery time from midazolam (p<0.01, *Figure 6I*). These results indicate that the recovery from midazolam is mediated predominantly by α1-R.

## The pro-recovery effect of activating LC$^{NE}$ neurons can be reversed by ICV injection or intra-VLPO microinjection of α1-R antagonist

To further determine the link between the LC and VLPO, we assessed the effect of photostimulation and pharmacology (*Figure 6J*). A mixture of the Cre-dependent AAV vector Ef1a-DIO-hChR2(H134R)-eYFP and mTh-Cre-AAV was injected bilaterally into the LC, and optical fibers were implanted (*Figure 6K*). Then, 2 weeks later, drug-delivery cannulas were implanted into the ICV or VLPO (*Figure 6M and O*). Then, we injected prazosin through cannulas within the ICV or VLPO to block α1-R and activate LC by photostimulation to further observe the effects of both on midazolam recovery time. The results showed that photostimulation of LC$^{NE}$ neurons could reduce the recovery time (p<0.05), whereas prazosin could prolong the recovery time (p<0.05), and that prazosin reversed the effects produced by LC activation (p<0.01, p<0.001, *Figure 6N and P*). Meanwhile, to further assess the effect of the above maneuvers on the depth of sedation, we performed EEG recordings. We found that cortical EEG activity was significantly reduced after LORR in all groups and remained at a low level during the LORR to RORR, while optogenetic activation of LC resulted in some high-amplitude EEG activity (*Figure 7A–C*, *Videos 7–9*), implying that LC$^{NE}$ activation may affect the depth of sedation.

A similar experiment was conducted utilizing the chemogenetic approach (*Figure 6Q*). The mixture of mTh-Cre-AAV and AAV-Ef1a-DIO-hM3Dq-mCherry viruses was injected into bilateral LC (*Figure 6R*); 2 weeks later, we determined the transfection efficiency of the chemogenetic virus in LC$^{NE}$ neurons in both male and female mice (*Figure 6S and X*). For male mice, bilateral drug-delivery cannulas were implanted into the ICV or VLPO (*Figure 6T and V*); for female mice, bilateral drug-delivery cannulas were implanted into the VLPO (*Figure 6Y*). α1-R antagonist was microinjected through the cannulas and behavioral tests were conducted as well as EEG recordings. Consistent with the above

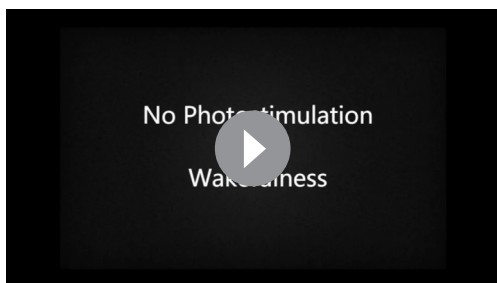

**Video 1.** Some of the main experimental videos throughout the study.

https://elifesciences.org/articles/97954/figures#video1

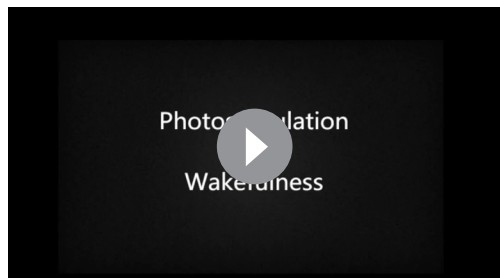

**Video 2.** Some of the main experimental videos throughout the study.

https://elifesciences.org/articles/97954/figures#video2

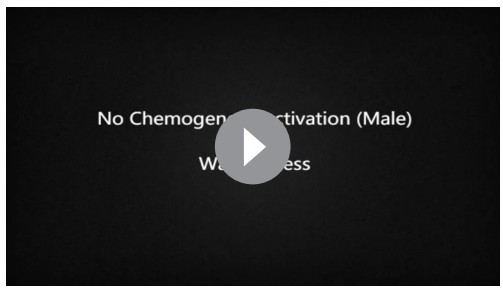

**Video 3.** Some of the main experimental videos throughout the study.
https://elifesciences.org/articles/97954/figures#video3

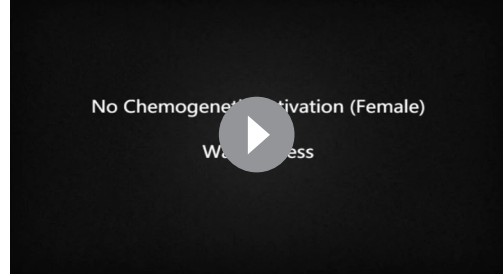

**Video 5.** Some of the main experimental videos throughout the study.
https://elifesciences.org/articles/97954/figures#video5

results, in male mice, both ICV and intra-VLPO injections of prazosin prolonged the recovery time (p<0.01, p<0.05), and reversed the pro-recovery effect produced by chemogenetic activation of LC$^{NE}$ neurons (p<0.01, *Figure 6U and W*). Also, in female mice, intra-VLPO injections of prazosin prolonged the recovery time (p<0.05) and reversed the pro-recovery effect produced by chemogenetic activation of LC$^{NE}$ neurons (p<0.001, *Figure 6Z*). In addition, chemogenetic activation of LC leads to some high-amplitude EEG activity (data from male mice, *Figure 7D–F*, *Videos 10–12*).

In summary, the recovery-promoting effects of all the aforementioned interventions that activate LC$^{NE}$ neurons can be reversed by prazosin. These findings further established the existence of a functional connection between LC$^{NE}$ neurons and VLPO that co-regulates recovery after midazolam. Notably, α1-R on the VLPO is the downstream target involved in the mechanism of recovery from midazolam.

## GABAA-R is an important mechanical binding site for midazolam in the LC

Midazolam is a short-acting central benzodiazepine receptor inhibitor that acts primarily on GABAA-R, thus increasing GABAergic neurotransmission and depressing the CNS. Hence, we further explored whether LC$^{NE}$ neurons exert their effects through GABAA-R after midazolam administration. We microinjected GCaMP6m into the LC in mice and implanted an optical fiber to observe changes in LC calcium signaling after the injection of gabazine, a GABAA receptor antagonist, into the lateral ventricle or LC (*Figure 8A*). We found that compared with the vehicle, ICV injection and intra-LC microinjection of 4 μg/mL gabazine could reduce the recovery time (p<0.05, *Figure 8C and E*). In addition, gabazine could increase the ΔF/F peak of calcium signaling in LC$^{NE}$ neurons before LORR and during the RORR to recovery phase (p<0.05, *Figure 8H–K*). These data indicate that GABAA-R in LC may be the initiating binding site of midazolam to produce altered consciousness.

We have already demonstrated that midazolam affects LC$^{NE}$ neuronal activity via GABAA-R in the LC, which then exerts its effects via downstream α1-R. First, drug-delivery cannulas were implanted into the LC and VLPO, respectively. One week later, gabazine was injected into the LC to block

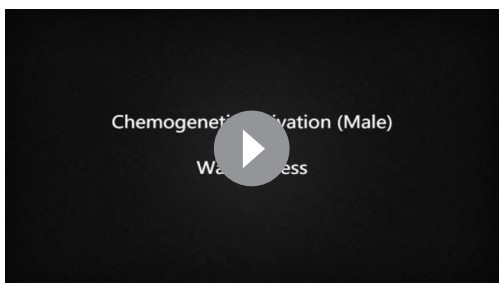

**Video 4.** Some of the main experimental videos throughout the study.
https://elifesciences.org/articles/97954/figures#video4

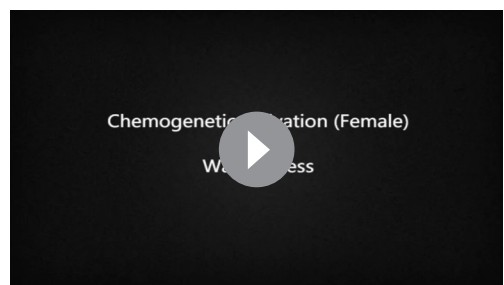

**Video 6.** Some of the main experimental videos throughout the study.
https://elifesciences.org/articles/97954/figures#video6



**Figure 4.** LC-VLPO NEergic neural circuit was involved in the recovery from midazolam. (**A**) Protocol for optogenetic or chemogenetic activation of LC[NE] neurons and recording of calcium signaling changes in VLPO. (**B**) A schematic of the location of optogenetic or chemogenetic virus injection site in the LC and calcium signaling recording virus injection site in the VLPO. (**C**) A representative image showing the co-expression of hChR2 and TH

*Figure 4 continued on next page*

*Figure 4 continued*

in the LC. (**D**) A representative image showing the co-expression of GCaMP6s and TH in the VLPO in male mice. (**E**) Quantification of the percentage of eYFP (+)/TH (+) colocalization in all TH (+) neurons in LC. (**F**) Quantification of the percentage of GCaMP6m (+)/TH (+) colocalization in all TH (+) neurons in VLPO. (**G**) Results show the effects of optogenetic activation of LC$^{NE}$ neurons on the recovery time of midazolam. (**H, I**) A representative image showing the co-expression of hM3Dq and TH in the LC of male and female mice. (**J**) Quantification of the percentage of mCherry (+)/TH (+) colocalization in all TH (+) neurons in LC. (**K**) A representative image showing the co-expression of GCaMP6s and TH in the VLPO in female mice. (**L**) Quantification of the percentage of GCaMP6m (+)/TH (+) colocalization in all TH (+) neurons in VLPO. (**M, N**) Results show the effects of chemogenetic activation of LC$^{NE}$ neurons on the recovery time of midazolam in male and female mice. (**O**) Schematic diagram of the method of recording the calcium signal at the VLPO, and sources of statistical data. (**P, Q**) Heatmap and statistical diagram of calcium signaling changes in bilateral VLPO at the after RORR stage with or without optogenetic activation of LC$^{NE}$ neurons (showing changes in calcium signaling during the RORR and the recovery period; optogenetic activation group, n = 3; no optogenetic activation group, n = 3). (The time points for RORR in mice were respectively selected as time = 0. The selected traces lasting for 20 s were based on the length of a complete Ca$^{2+}$ signal.) (**R**) The peak ΔF/F in the VLPO with or without activation of LC$^{NE}$ neurons using optogenetics. (**S–T, V–W**) Heatmap and statistical diagram of calcium signaling changes in bilateral VLPO at the after RORR stage with or without chemogenetic activation of LC$^{NE}$ neurons in male and female mice (showing changes in calcium signaling during the RORR and the recovery period; chemogenetic activation group, n = 3; no chemogenetic activation group, n = 3). (**U, X**) The peak ΔF/F in the VLPO with or without activation of LC$^{NE}$ neurons using chemogenetics in male and female mice. LC: locus coeruleus; VLPO: ventrolateral preoptic nucleus; i.p.: intraperitoneal injection; LORR: loss of righting reflex; RORR: recovery of righting reflex; TH: tyrosine hydroxylase; PS: photostimulation; *p<0.05; **p<0.

---

GABAA-R, while prazosin was injected into the VLPO to antagonize α1-R (*Figure 8L and M*). We found that microinjection of gabazine into the LC could reduce the recovery time (p<0.05), whereas microinjection of prazosin into the VLPO could increase the recovery time (p<0.05, *Figure 8O*). In addition, we found that microinjection of prazosin into VLPO prolonged the recovery time and significantly reversed the blockade of LC GABAA-R-induced pro-recovery effect (p<0.05, p<0.001, *Figure 8O*).

To investigate the role of GABAA-R on LC$^{NE}$ neurons in the modulation of recovery from midazolam, we performed a loss-of-function study using shRNA to knock down GABAA-R in the bilateral LC$^{NE}$ neurons. As assessed by immunofluorescence staining analysis, the expression of GABAA-R on LC$^{NE}$ neurons was significantly reduced compared to the sham group (p<0.0001, *Figure 9A–F*). We found that knockdown of GABAA-R prolonged the recovery time from midazolam, while it did not affect the dose of midazolam-induced LORR (p<0.01, *Figure 9G–I*, *Videos 13 and 14*).

## GABAergic and NEergic systems interact with each other and co-regulate the recovery from midazolam

In previous experiments, we preliminarily proved that GABAergic and NEergic systems co-regulate recovery from midazolam by the pharmacological method. Then, we combined the gene knockdown technique with fiber photometry to delve deeper (*Figure 9J–Q*). We found that the ΔF/F peak of calcium signaling in the VLPO NEergic terminals in the shRNA group was significantly reduced compared with the sham group (p<0.05, *Figure 9R–T*).

More importantly, we knocked down GABAA-R on LC$^{NE}$ neurons, resulting in fewer targets for midazolam and weakened function of the GABAergic system, and similar to previous results, recovery time from midazolam was reduced. Meanwhile, microinjecting NE α1-R antagonist in the VLPO, we found that the reduction of recovery time caused by the knockdown of GABAA-R was reversed (p<0.05, *Figure 9Z*, *Videos 15 and 16*), suggesting that the NEergic system is indeed involved in recovering from midazolam with NE α1-R on VLPO neurons is an important functional site in the LC-VLPO neural circuit.

## Discussion

Midazolam holds significant potential in the treatment of psychiatric disorders as sedative hypnotics. Although the number of clinical studies addressing the effects of midazolam on consciousness as well as behavior is gradually increasing, the specific neural mechanisms by which it alters consciousness remain unclear. To fill this gap, we opted to investigate midazolam, a clinical drug, to study in mice to

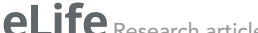

**Figure 5.** Activation of LC-VLPO NEergic neural circuit promotes recovery from midazolam. (**A**) Protocol for intra-LC/VLPO microinjection of the virus and optogenetic activation of VLPO. (**B, C**) Schematic of LC-VLPO long-range optogenetic activation and the location of optogenetic virus microinjection and optic fiber implantation. (**D**) Representative images showing the co-expression of hChR2 and TH in the LC. (**E**) Quantification of the percentage of eYFP (+)/TH (+) colocalization in all TH (+) neurons in LC. (**F**) Distributions of hChR2-eYFP terminals from LC-NA neurons and DAPI labeling in the VLPO. (**G**) Results show the effect of LC-VLPO long-range optogenetic activation on the recovery time of midazolam. (**H**) Protocol for intra-LC/VLPO microinjection of the virus and chemogenetic activation of VLPO. (**I, J**) Schematic of LC-VLPO long-range chemogenetic activation and the location of chemogenetic virus microinjection and cannula implantation. (**K**) Representative images showing the co-expression of mCherry and TH in the LC. (**L**) Quantification of the percentage of mCherry (+)/TH (+) colocalization in all TH (+) neurons in LC. (**M**) Distributions of hM3Dq-mCherry terminals from LC-NA neurons and DAPI labeling in the VLPO. (**N**) Results show the effect of LC-VLPO long-range chemogenetic activation on the recovery time of midazolam in male and female mice. LC: locus coeruleus; VLPO: ventrolateral preoptic nucleus; i.p.: intraperitoneal injection; LORR: loss of righting reflex; RORR: recovery of righting reflex; TH: tyrosine hydroxylase; PS: photostimulation; *p<0.05; **p<0.01.

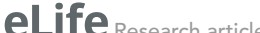

**Figure 6.** ICV injection or intra-VLPO microinjection of α1-R antagonist reverses the pro-recovery effect of optogenetic or chemogenetic activation of LC^NE neurons. (**A**) A schematic of ICV injection of different noradrenergic receptor agonists and antagonists. (**B**) A representative photomicrograph showing the tracks of cannulas implanted into the lateral ventricle. (**C–H**) Results show the effects of ICV injection of different doses of

*Figure 6 continued on next page*

*Figure 6 continued*

phenylephrine, prazosin, clonidine, yohimbine, isoprenaline, and propranolol on the recovery time of midazolam. (**I**) Results show the effects of ICV injection of phenylephrine (20 mg/mL) and different doses of propranolol on the recovery time of midazolam. (**J**) Protocol for PS of LC$^{NE}$ neurons and ICV injection or intra-VLPO microinjection of α1-R antagonist. (**K**) Schematic of the brief process of optogenetic activation. (**L**) The representative photomicrograph showing microinjection and optical fiber locations of the co-expression of hChR2 and TH in the LC. Moreover, quantification of the percentage of eYFP (+)/TH (+) colocalization in all TH (+) neurons in LC is shown. (**M, O**) A representative photomicrograph showing the tracks of cannulas implanted into the lateral ventricle and bilateral VLPO. (**N, P**) Results show the effects of PS of LC$^{NE}$ neurons and ICV injection or intra-VLPO microinjection of prazosin on the recovery time of midazolam. (**Q**) Protocol for chemogenetic activation of LC$^{NE}$ neurons and ICV injection or intra-VLPO microinjection of an α1-R antagonist. (**R**) Schematic of the location of virus injection. (**S**) The representative photomicrograph showing the co-expression of hM3Dq and TH in the LC of male mice. Moreover, quantification of the percentage of mCherry (+)/TH (+) colocalization in all TH (+) neurons in LC is shown. (**T, V**) A representative photomicrograph showing the tracks of cannulas implanted into the lateral ventricle and bilateral VLPO in the male mice. (**U, W**) Results show the effects of chemogenetic activation of LC$^{NE}$ neurons and ICV injection or intra-VLPO microinjection of prazosin on the recovery time of midazolam in male mice. (**X**) A representative photomicrograph showing the co-expression of hM3Dq and TH in the LC. Moreover, quantification of the percentage of mCherry (+)/TH (+) colocalization in all TH (+) neurons in LC is shown. (**Y**) The tracks of cannulas implanted into the bilateral VLPO in the female mice. (**Z**) Results show the effects of chemogenetic activation of LC$^{NE}$ neurons and intra-VLPO microinjection of prazosin on the recovery time of midazolam in female mice. LC: locus coeruleus; VLPO: ventrolateral preoptic nucleus; i.p.: intraperitoneal injection; LORR: loss of righting reflex; RORR: recovery of righting reflex; TH: tyrosine hydroxylase; ICV: intracerebroventricular; PS: photostimulation; *p<0.05; **p<0.01; ***p<0.001.

find a promising target for the development of an effective strategy to provide targeted therapies for the reversal of midazolam-induced delayed recovery.

In the current study, we employed a cascade approach, initially utilizing pharmacology and enzyme-linked immunosorbent assay (ELISA), to validate that the central NE system plays an important role in recovery from midazolam sedation. We found that midazolam administration significantly reduced NE content in the brainstem, whereas IP injection of atomoxetine, an NE reuptake inhibitor, could shorten recovery time from midazolam. Conversely, peripheral injection of DSP-4 reverses the pro-recovery effect of atomoxetine. Second, based on calcium signaling, we found a decrease in LC$^{NE}$ neuronal activity after midazolam administration. Concordantly, optogenetic and chemogenetic activation of LC$^{NE}$ neurons could facilitate recovery from midazolam, whereas elective ablation of LC$^{NE}$ neurons by intra-LC microinjection of DSP-4 attenuates recovery. Third, we turned our attention to the downstream pathway of the LC. We found that optogenetic and chemogenetic activation of LC$^{NE}$ neurons could influence the activity of VLPO, and ChR2 can be transmitted from the LC to the VLPO across synapses, suggesting that the LC can project to the VLPO. Next, ICV injection of different adrenergic receptor agonists or antagonists suggested the main involvement of α1-R in the regulative effect of recovery from midazolam. Also, we microinjected α1-R antagonist into VLPO and found that the pro-arousal effect induced by activation of LC$^{NE}$ neurons was reversed, suggesting that α1-R in the VLPO is more responsive to regulate recovery from midazolam. Then, ICV injection of the GABAA-R antagonist gabazine or microinjection of gabazine into the LC shortened the recovery time from midazolam, and we found that antagonizing GABAA-R could inhibit the activity of LC, suggesting that GABAA-R in the LC may be an initiating target for the action of midazolam. Finally, by using pharmacological method and gene knockdown technique we found that GABAergic and NEergic systems co-regulated recovery from midazolam, and GABAA-R on LC$^{NE}$ neurons and NE α1-R on VLPO were the key points. More importantly, in this study, both male and female mice were used in all chemogenetic activation experiments, and we found that sex discrepancies did not affect the results of the experiment, which signified that the intervention that promoted rebooting from midazolam-induced unconsciousness was equally effective in both male and female mice. Our findings have a universal applicability as well as great clinical translational significance. For the dopamine-beta-hydroxylase (Dbh) promotor applied in the optogenetic and chemogenetic studies, as supported by the literature, Dbh, located in the inner membrane of noradrenergic and adrenergic neurons, is an enzyme that catalyzes the conversion of dopamine to NE, and therefore plays an important role in noradrenergic neurotransmission. Dbh is a marker of noradrenergic neurons. *Zhou et al., 2020* clarified the probe specifically labeled noradrenergic neurons by immunolabeling for Dbh. Recently, Dbh promoter has been used in several



**Figure 7.** Effect of ICV injection of α1-R antagonist and activation of LC[NE] neurons on EEG activity. (**A, D**) Schematic diagram of EEG recording method. (**B**) EEG and spectrum of mice in vehicle + no PS, vehicle + PS, prazosin + no PS, and prazosin + PS at the *Before LORR*, *LORR-RORR*, and *After RORR* stages. (**C**) Alpha, Beta, Theta, Gamma, and Delta wave proportion of EEG in four groups of mice in different stages. (**E**) EEG and spectrum

*Figure 7 continued on next page*

*Figure 7 continued*

of mice in vehicle + no CNO, vehicle + CNO, prazosin + no CNO, and prazosin + CNO at the *Before LORR, LORR-RORR*, and *After RORR* stages. (**F**) Alpha, Beta, Theta, Gamma, and Delta wave proportion of EEG in four groups of mice in different stages. LC: locus coeruleus; LORR: loss of righting reflex; RORR: recovery of righting reflex; PS: photostimulation. *p<0.05; **p<0.01; ***p<0.001.

studies (e.g., *Han et al., 2024*; *Lian et al., 2023*). The Dbh-Cre mice are widely used to specifically labeled noradrenergic neurons (e.g., *Li et al., 2023*; *Breton-Provencher et al., 2022*; *Liu et al., 2024*). It is difficult to distinguish the role of NE or DA neurons when using the Th promoter in VLPO. Therefore, we used Dbh promoter with more specific labeling. LC is the main noradrenergic nucleus of the central nervous system. In our study, we injected rAAV-Dbh-GCaMP6m-WPRE (*Figures 2 and 8*) and rAAV-Dbh-EGFP-S'miR-30a-shRNA GABAA receptor–3'-miR30a-WPRES (*Figure 9*) into the LC. The results showed that Dbh promoter could specifically label noradrenergic neurons in the LC, while nonspecific markers outside the LC were almost absent.

Altered states of consciousness attributed to the action of sedative drugs on molecular targets in the CNS have been the focus of research over the past decades. GABAA-R is a well-recognized target of different sedative agents. GABAA-R is a ligand-gated Cl- channel that mediates the majority of the fast inhibitory neurotransmission in the CNS and has a crucial role in regulating brain excitability. The GABAA-R family comprises 19 subunit genes, including α1–6, β1–3, γ1–3, δ, ε, θ, π, and $\rho$ 1–3, which assemble to form a pentameric structure (*Ghit et al., 2021*). The GABAA-R has a GABA-binding receptor site and some regulatory sites for binding of various substances, and the best characterized regulatory site is the benzodiazepine (BZ) one (*Sigel and Ernst, 2018*). Midazolam is a prevalent psychotropic substance that acts as a positive allosteric modulator of this receptor (*Olsen, 2018*). The drug binds highly specifically to the BZ-binding site on the GABAA-R complex, modulating the affinity of GABA and GABA-binding receptor sites and reducing the excitability of the CNS, thus inducing sedation, hypnosis, and unconsciousness. Receptors containing the α1, 2, 3, 5, and γ2 subunit interface form the BZ-binding site of GABAA-R, suggesting the significance of these subunits in midazolam-mediated effects (*Maramai et al., 2020*). Notably, β3-knockout mice exhibit a shorter duration of LORR in response to midazolam, suggesting a role of β3-containing GABAA-R in mediating midazolam-induced unconsciousness (*Quinlan et al., 1998*). According to our findings, midazolam modulates downstream neural pathways by initially acting on GABAA-R in the LC. It is future studies of membrane proteins and ion channels in LC or non-LC NEergic neurons that we call for to fully understand the molecular and neural circuit targets of sedative-hypnotics.

Midazolam elicits a de-consciousness state resembling natural sleep, likely involving endogenous sleep-arousal neural nuclei and circuits to induce hypnosis and loss of consciousness. In conjunction with past studies, midazolam and other sedative drugs act primarily by enhancing GABAergic inhibitory neurotransmission, implying a sleep-like facilitating effect of the GABAergic system. However, the mechanism of neural circuits downstream of GABA that regulate changes in consciousness remains elusive.

Arousal in the brain depends on many areas that can simultaneously receive diverse signals, with the ascending reticular activating system (ARAS) overseeing arousal and wakefulness maintenance (*Lee and Dan, 2012*). LC[NE] neurons, which are an important component of the ARAS, reportedly

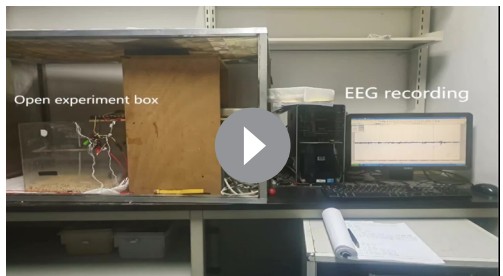

**Video 7.** Some of the main experimental videos throughout the study.

https://elifesciences.org/articles/97954/figures#video7

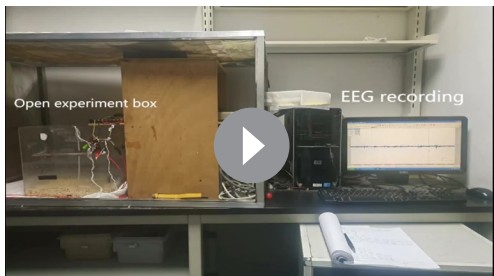

**Video 8.** Some of the main experimental videos throughout the study.

https://elifesciences.org/articles/97954/figures#video8

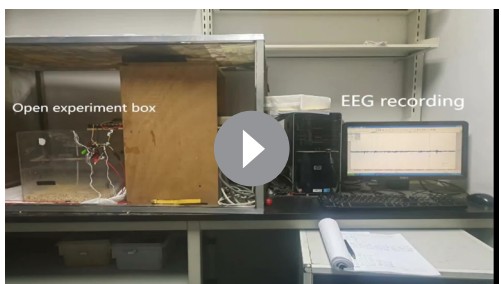

**Video 9.** Some of the main experimental videos throughout the study.
https://elifesciences.org/articles/97954/figures#video9

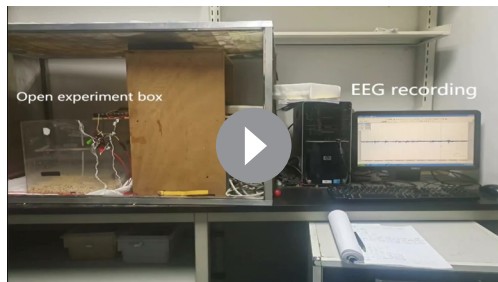

**Video 11.** Some of the main experimental videos throughout the study.
https://elifesciences.org/articles/97954/figures#video11

play a key role in the maintenance of arousal and alertness (*Berridge et al., 2012*). Specifically, we posit that LC^NE neurons and their associated circuits are involved in modulating recovery from midazolam. In this study, LC^NE neurons were activated by photostimulation and chemogenetics to rapidly induce the transition from unconsciousness to awakening. As a pivotal arousal center, the LC receives arousal-related inputs and extensively innervates the cerebral cortex and forebrain structures with NEergic projections (*Hayat et al., 2020*). We initially pharmacologically validated the role of the central NEergic system in modulating midazolam recovery. Subsequently since the LC is the predominant nucleus in the brain for the release of NE, we further demonstrated the role of LC^NE neurons in midazolam recovery through selective activation and degradation of them. Here, our study elucidates the critical role of NEergic neurons in the LC in midazolam recovery by pharmacology, optogenetics, and chemogenetics.

Given that the VLPO lies downstream of the LC and plays a crucial role in sleep-related information integration, we spotlighted the LC-VLPO neural circuit. The VLPO harbors diverse neuronal populations, including GABAergic and galaninergic neurons, which are recognized as sleep-active neurons (*Arrigoni and Fuller, 2022*). Intriguingly, there is another cluster of glutamatergic neurons in the VLPO that promote arousal (*Chung et al., 2017*; *Vanini et al., 2020*). A recent report showed that LC^NE neurons synergistically facilitate arousal from sleep by simultaneously activating wake-active neurons and inhibiting sleep-active neurons in the VLPO. This regulation of the two types of neurons in the VLPO is mediated by distinct adrenergic receptors (*Liang et al., 2021*). However, our study differs significantly. First, we demonstrate that LC may play an irreplaceable role in midazolam-induced loss of consciousness because atomoxetine is a reuptake inhibitor of widespread NE in the brain, but our selective disruption of LC rendered its effect lost. Second, our study pinpoints GABAA-R in the LC as a potential key target for midazolam's effects. Third, we identified this α1 receptor as a necessary target through pharmacology and further validated it with more specific means. Finally, we found that intra-VLPO microinjection of the α1 receptor antagonist prazosin significantly reversed the pro-recovery effect of intra-LC microinjection of GABAA receptor antagonist or activation of LC^NE neurons. Crucially, atomoxetine, a psychotropic drug for treating attention-deficit hyperactivity disorder (ADHD), is a promising candidate for translational research to prevent delayed recovery from midazolam abuse in

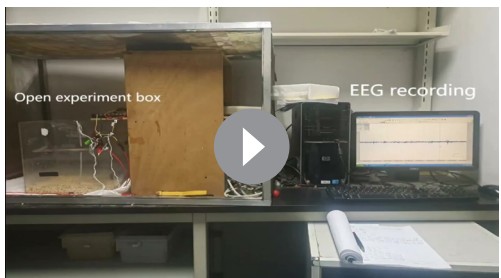

**Video 10.** Some of the main experimental videos throughout the study.
https://elifesciences.org/articles/97954/figures#video10

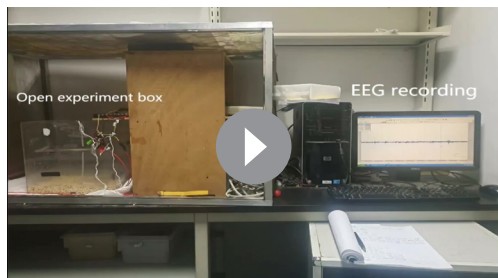

**Video 12.** Some of the main experimental videos throughout the study.
https://elifesciences.org/articles/97954/figures#video12



**Figure 8.** GABAA-R is an important mechanical binding site for midazolam in the LC. (**A**) Protocol for the fiber optic recording of calcium signals in the LC after ICV injection or intra-LC microinjection of different doses of gabazine. (**B, D**) A representative photomicrograph showing the tracks of cannulas implanted into the lateral ventricle and the bilateral LC. (**C, E**) Results show the effects of ICV injection or intra-LC microinjection of different doses of gabazine on the recovery time of midazolam. (**F**) A representative photomicrograph showing the microinjection and optical fiber locations and the co-expression of GCaMP6s and TH in the LC. (**G**) Quantification of the percentage of GCaMP6m (+)/TH (+) colocalization in all TH (+) neurons in LC. (**H**) Schematic diagram of the method of recording the calcium signal at the LC and sources of statistical data. (**I**) A heatmap of calcium signaling changes in bilateral LC^NE neurons induced by midazolam with or without ICV injection of gabazine. (*Vehicle group*: n = 2; *gabazine group*: n = 2; calcium signaling data from two representative mice per group. Calcium signaling information of each mouse in the stage *Before RORR* and *R-Recovery* were selected for presentation and further statistical analysis. The time points for midazolam injection and RORR in mice were respectively selected as time = 0. The selected traces lasting for 20 s were based on the length of a complete Ca²⁺ signal.) (**J**) A statistical diagram of calcium signaling changes in bilateral LC with or without ICV injection of gabazine. (**K**) The peak ΔF/F in the *Before RORR* and *RORR-Recovery* stages with or without ICV injection of gabazine. (**L**) Protocol for intra-LC microinjection of GABAA-R antagonist and intra-VLPO microinjection of α1-R antagonist. (**M**) A schematic of the

*Figure 8 continued on next page*

*Figure 8 continued*

position of cannula implantation. (**N**) A representative photomicrograph showing the tracks of cannulas implanted into the bilateral VLPO. (**O**) Results show the effects of intra-LC injection of gabazine and intra-VLPO injection of prazosin on the recovery time of midazolam. LC: locus coeruleus; VLPO: ventrolateral preoptic nucleus; i.p.: intraperitoneal injection; LORR: loss of righting reflex; RORR: recovery of righting reflex; TH: tyrosine hydroxylase; ICV.: intracerebroventricular; *p<0.05; **p<0.01; ***p<0.001.

patients with ADHD. Essentially, targeting the central α1-R, particularly in VLPO wake-active neurons, may yield rational therapeutic strategies to prevent delayed recovery. Notably, the VLPO in turn sends GABAergic inhibitory projections to various wake-promoting nuclei throughout the neuroaxis, including the LC (*Saper et al., 2005*). Studies indicate that the activated VLPO releases GABA to the LC, thus inducing NREM sleep, while NE released from the activated LC inhibits VLPO sleep-active neuron activity and promotes wakefulness (*Brown et al., 2012*; *Van Egroo et al., 2022*). As such, a deeper exploration of the interactions between diverse VLPO neuronal populations and their specific connections with the LC is warranted in our study.

The translational implications of a study are indeed a crucial measure of its significance. Our current study has the potential to translate into guidance for the clinical use of midazolam and allow for reducing unwanted drug side effects. On the one hand, we have selected the clinical drug midazolam for animal experiments, which will help us develop useful drugs to treat midazolam-induced serious complications (e.g., delayed recovery, ventilator-associated pneumonia, and coma) in the future, thus further advancing clinical trials. It provides some hints for managing agitated patients with neuro-psychiatric disorders. On the other hand, in addition to intravenous or intramuscular administration, midazolam can also be taken orally to treat insomnia, ameliorate anxiety, and treat seizures and status epilepticus (*Conway et al., 2016*; *Nordt and Clark, 1997*; *Smith and Brown, 2017*). Oral midazolam affects consciousness to a greater or lesser extent, and some suicidal patients have even suffered immediate death from massive doses of midazolam. Our study thus offers valuable insights for medication guidance in such cases.

In summary, our study demonstrates that, in mice, the LC-VLPO NEergic neural circuit contributes to the rapid modulation of altered consciousness induced by midazolam (*Figure 10*), which is essential to ameliorate the complications of its abuse. It will pinpoint the relevant regions involved in response to midazolam and provide a perspective to help elucidate the dynamic changes of neural circuits in the brain during altered consciousness. It suggests a promising new avenue of investigation for the targets of timely recovery from midazolam-induced loss of consciousness, thereby minimizing the risk of delayed recovery and associated complications.

## Limitations

This study shows that the LC-VLPO NEergic neural circuit plays an important role in modulating midazolam recovery. However, the specificity of LC NE downstream neurons in the VLPO is not explained in this paper, which is our next research direction, VLPO neurons and their downstream regulatory mechanisms may be involved in other nervous systems except the NE nervous system, and the deeper and more complex mechanisms need to be further investigated. In addition, given the pharmacological actions of midazolam, other areas may also be involved. Current studies suggest that the neural network involved in the recovery of consciousness consists of the prefrontal cortex, basal forebrain, brain stem, hypothalamus, and thalamus (*Nguyen and Postnova, 2021*). The role of these regions in midazolam recovery remains to be further investigated. Therefore, we will apply more specific experimental methods to determine the importance of LC-VLPO NEergic neural circuit and related NE receptors in the midazolam recovery, and conduct further studies on other relevant brain neural regions, hoping to more fully elucidate the mechanism of midazolam recovery in the future.

## Methods
### Animals

The experimental animals in this study were male and female wild-type C57BL/6J mice purchased from the Animal Experiment Center of Zhejiang University. All mice were housed and bred in the SPF-Class House in a standard condition (indoor temperature 25°C, ambient humidity 65%, 12 hr/12 hr light/dark cycle) with rodent food and water available ad libitum. All mice were aged 8 weeks and



**Figure 9.** GABAergic and NEergic systems interact with each other, co-regulate the recovery from midazolam.
(**A**) Protocol for exploring the influence of knocking down the GABAA-R on the recovery time from midazolam.
(**B, C**) Schematic diagram and the representative photomicrograph showing the position of virus injection.
(**D**) A representative photomicrograph showing the co-expression of shRNA, TH, and GABAA-R in the LC. (**E**)

*Figure 9 continued on next page*

*Figure 9 continued*

Representative photomicrograph of the difference of GABAA-R expression in the LC after shRNA-mediated knockdown. (**F**) The quantification of GABAA-R (+)/TH (+) cells in the LC between the shRNA group and the sham group. (**G**) Number of LORR and no LORR induced by different doses of midazolam in GABAA-R knockdown mice. (**H**) Rate of LORR (%) in GABAA-R knockdown mice at different doses of midazolam. (**I**) The statistical bar shows shRNA-mediated GABAA-R knockdown effect on the recovery time from midazolam. (**J, K**) Protocol for exploring the calcium signals changes in the VLPO after knocking down the GABAA-R. (**L, M**) Schematic diagram and the representative photomicrograph showing the position of virus injection and the co-expression of shRNA, TH, and GABAA-R in the LC. (**N, O**) Schematic diagram of the brief process of calcium signal recording and the representative photomicrograph showing the co-expression of GCaP6m and TH in the VLPO. (**P**) Quantification of the percentage of GCaMP6m (+)/TH (+) colocalization in all TH (+) neurons in VLPO (**Q**) Schematic diagram of the method of recording the calcium signal at the VLPO and sources of statistical data. (**R, S**) Heatmap and statistical diagram of calcium signaling changes in bilateral VLPO at the after RORR stage with or without knocking down GABAA-R on LC$^{NE}$ neurons (showing changes in calcium signaling during the RORR and the recovery period; shRNA group, n = 3; sham group, n = 3). (The time points for RORR in mice were respectively selected as time = 0. The selected traces lasting for 20 s were based on the length of a complete Ca$^{2+}$ signal.) (**T**) The peak ΔF/F in the VLPO with or without knocking down GABAA-R on LC$^{NE}$ neurons. (**U**) Protocol for exploring the effect of blocking α1-R in the VLPO and knocking down the GABAA-R in the LC on the recovery time from midazolam. (**V, W**) Schematic diagram and the representative photomicrograph showing the positions of cannula implantation. (**X, Y**) Schematic diagram and the representative photomicrograph showing the position of virus injection and the co-expression of shRNA, TH, and GABAA-R in the LC. (**Z**) Results show recovery time between sham, shRNA + vehicle, and shRNA + prazosin. LC: locus coeruleus; VLPO: ventrolateral preoptic nucleus; i.p.: intraperitoneal injection; LORR: loss of righting reflex; RORR: recovery of righting reflex; TH: tyrosine hydroxylase; *p<0.05; ***p<0.001; ****p<0.0001.

weighed 22–25 g at the start of the experiments. All experiments were mainly performed between 9:00 and 16:00 (*Supplementary file 1*).

## Establishment of a mice model of midazolam administration

We randomly divided healthy C57BL/6J mice into four groups (n = 8). These mice were used to determine the IP dose of midazolam that would achieve the optimal depth without inhibiting respiration before starting the formal experiment. Finally, we identified the lowest effective dose that was 100% successful in inducing loss of consciousness in mice, that is, 60 mg/kg midazolam (H-19990027, Jiangsu Nhwa Pharmaceutical Co, Ltd), as the optimal dosage for IP administration in mice and used this dose in subsequent experiments.

## Evaluation of recovery time

Before the experiment, the bottom of the open box with no lid and air circulation was covered with cotton pads and an electric blanket was placed on the lower side of the open box, and the temperature in the open box was preheated for 10 min before the experiment to keep the temperature in the open box suitable and constant during the experiment. Eight-week-old C57BL/6J mice were placed in the open box for 30 min, and then placed in the open box again after IP injection of the appropriate dose of midazolam (60 mg/kg) to start the timer. At the same time, the open box was gently

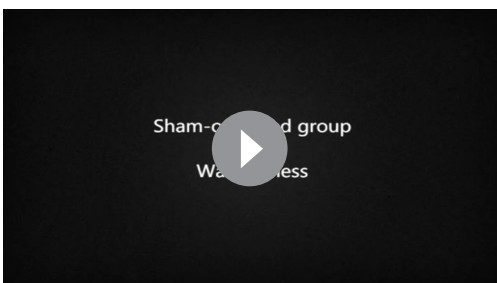

**Video 13.** Some of the main experimental videos throughout the study.
https://elifesciences.org/articles/97954/figures#video13

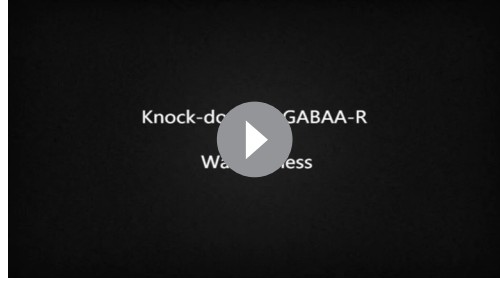

**Video 14.** Some of the main experimental videos throughout the study.
https://elifesciences.org/articles/97954/figures#video14

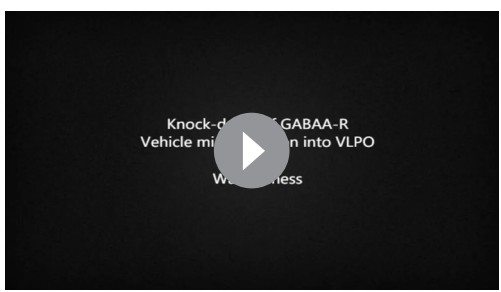

**Video 15.** Some of the main experimental videos throughout the study.
https://elifesciences.org/articles/97954/figures#video15

rotated every 15 s. When the mice were rotated to the dorsal recumbent position and could not be voluntarily turned over for more than 1 min, it was the LORR, that is, loss of consciousness from midazolam. Subsequently, the mice were placed in the dorsal recumbent position for continuous observation, and when the mice could autonomously return to the normal position from the dorsal recumbent position, they were considered RORR. The time from the onset of LORR to RORR was recorded as the recovery time of midazolam.

## ELISA

The tyrosine hydroxylase (TH) content and specific enzyme activity were measured in two groups of C57BL/6J mice (n = 6 per group), one group was given 60 mg/kg midazolam intraperitoneally and the other group was not. The whole brain was removed and sectioned, and then prosencephalon and brainstem tissue samples were collected separately, and TH levels in these tissues were measured using an ELISA kit according to the manufacturer's instructions (YS-M195, YS-M195-1, ELISA Kit, Yan Sheng Biological Technology Co, Ltd, Shanghai, China). The optical density (OD) was measured at 450 nm and 630 nm using an ELISA microplate reader (SynergyMax M5, iD5, Molecular Devices, San Jose, CA) (***Supplementary file 1***).

## Stereotaxic surgery and virus microinjection surgery

Eight-week-old C57BL/6J mice were weighed and injected intraperitoneally with 30 mg/kg of pentobarbital and placed in the rearing cage for about 5–10 min. After complete anesthesia, the mice were fixed on a stereotactic apparatus (68018, RWD Life Sciences, Shenzhen, China) and the head hairs of the mice were shaved off with a razor. A heating pad was used to maintain the body temperature of mice at 37°C during the procedure and an ophthalmic ointment was applied to their eyes to avoid dryness. To expose the skull, the scalp was sterilized and a longitudinal incision was made. The periosteum was then removed with 3% hydrogen peroxide and the residue was rinsed with saline. After drilling holes in the skull, the virus was injected into the target site at a flow rate of 40 nL/min via an ultramicro pump (160494 F10E, WPI). We left the needle at the target site for 10 min after injection to allow for diffusion of the virus. To target the LC, the coordinates were AP: −5.41 mm; ML: ±0.9 mm; DV: −3.8 mm. To target the VLPO, the coordinates were AP: −0.01 mm; ML: ±0.65 mm; DV: −5.7 mm. To target the lateral ventricular, the coordinates were AP: −0.47 mm, ML: −1.00 mm, DV: −2.40 mm. According to the experimental needs, optical fibers or catheters were subsequently implanted in the corresponding location and secured with dental cement. After that, the incision was sutured and the mice were placed on a heating blanket to be transferred to a cage when the mice were awake (***Supplementary file 1***).

## Optogenetics

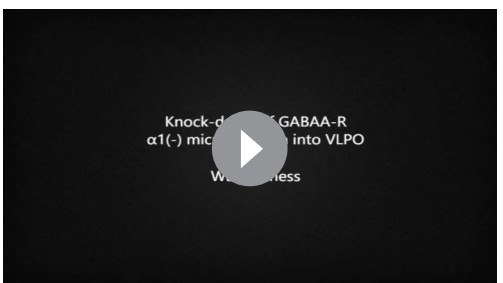

**Video 16.** Some of the main experimental videos throughout the study.
https://elifesciences.org/articles/97954/figures#video16

Three weeks before the experiment began, the LC or VLPO was microinjected with an optogenetic virus of 100 nL of a 1:3 volume mixture of mTh-Cre-AAV/Dbh-Cre and AAV-EF1a-DIO-hChR2(H134R)-eYFP (viral titer: 5.00 × 10^12 vg/mL, Brain VTA Technology Co, Ltd, Wuhan, China). One week before the experiment began, the LC or VLPO was implanted with optical fibers (FOC-W-1.25-200-0.37-3.0, Inper, Hangzhou, China). Th-hChR2 expression in the LC and VLPO was determined by immunohistochemistry after the completion of optogenetic experiments. The mice were intraperitoneally injected with



**Figure 10.** Summary of neural mechanism of rebooting the unconsciousness caused by midazolam. By using pharmacology, optogenetics, chemogenetics, fiber photometry, and gene knockdown, we revealed the role of LC-VLPO noradrenergic neural circuit in regulating midazolam-induced altered consciousness. This effect was mediated by α1 adrenergic receptors. Moreover, gamma-aminobutyric acid receptor type A (GABAA-R) is a mechanistically important binding site in the LC for midazolam. RISC: RNA-induced silencing complex; LC: locus coeruleus; VLPO: ventrolateral preoptic nucleus; NE: norepinephrine.

midazolam, and 20 min later, they were given 465 nm blue light to activate neurons in the LC or VLPO. In this study, we used different parameters for optogenetic activation: 20 min/2 mW; 20 min/4 mW; 10 min/4 mW; and 20 min/4 mW. We finally identified 20 min/4 mW as the optimal optogenetic activation parameter and used it in the subsequent experiments.

## Chemogenetics

Three weeks before starting the experiment, chemogenetic viruses of 100 nL of a 1:3 volume mixture of mTh-Cre-AAV/Dbh-Cre and pAAV-EF1a-DIO-hM3D-mCherry (viral titer: 5.00 × 10¹² vg/mL, Brain VTA Technology Co, Ltd) were microinjected into the LC or VLPO, and Th-hM3Dq expression in the LC and VLPO was determined by immunohistochemistry after the completion of chemogenetic experiments. Clozapine-N-oxide (CNO) (HY-17366, MedChemExpress) was dissolved in saline, and the mice were intraperitoneally injected with CNO 20 min after intraperitoneally injected with midazolam to activate neurons using the chemogenetics approach. In this study, we tested two concentrations of

CNO: 0.1 mg/kg and 0.2 mg/kg. We concluded that 0.2 mg/kg was the optimal concentration and used this concentration for the subsequent IP administration.

## Fiber photometry

We microinjected 100 nL of rAAV-DβH-GCaMP6m-WPRE-hGH pA (viral titer: $2.00 \times 10^{12}$ vg/mL, Brain VTA Technology Co, Ltd) into the LC or VLPO. Two weeks later, an optical fiber (FOC-W-1.25-200-0.37-3.0, Inper) was implanted in the same area located 0.05 mm above the virus injection point (AP: −5.41 mm; ML: ±0.9 mm; DV: −3.75 mm/AP: −0.01 mm; ML: ±0.65 mm; DV: −5.65 mm) and fixed in place with dental cement. Then, 3 weeks after viral expression and 1 week after fiber optic placement, the calcium activity of the target neurons was monitored. Fluorescence emissions were recorded using a fiber photometry system (Inper, C11946) using a 488 nm diode laser, and the mice were placed in a dark box to avoid the influence of natural light source on calcium signal recording and accessed to the computer recording software to record calcium signals. After the recording was completed, the results recorded by the optical fiber were read into the calcium signaling data analysis software. The calcium signalings were recorded 30 min before LORR when the mice were awake. Then, midazolam was injected intraperitoneally. After the RORR, the experiment was terminated after 10 min of optical fiber recording. In *Figure 2*, the time points for awake, midazolam injection, LORR and RORR in mice were respectively selected as time = 0, while in *Figure 4*, RORR in mice was selected as time = 0. The selected traces lasting for 20 s were based on the length of a complete $Ca^{2+}$ signal. The calcium signal intensity was calculated as the value of ΔF/F = (F-F0)/F0.

## GABAA-R knockdown

We microinjected 80 nL of rAAV-Dbh-EGFP-S'miR-30a-shRNA (GABAA receptor)–3'-miR30a-WPRES (viral titer: 5E+12 vg/mL, Brain VTA Technology Co, Ltd) into the LC. Four weeks later, these mice were sacrificed and perfused for counting the GABAA-R (+) and tyrosine hydroxylase-positive (TH+) cells to evaluate the effect of GABAA-R knockdown.

## Pharmacological experiments

### Effects of IP injection of atomoxetine and DSP-4 and their interaction on the recovery time of midazolam

Atomoxetine (Ca# Y0001586, Sigma-Aldrich), which selectively inhibits presynaptic uptake of NE and enhances NE function, was dissolved in saline. One hour before IP injection with midazolam, C57BL/6J mice received IP injection of atomoxetine (10 mg/kg or 20 mg/kg) or saline (vehicle). To evaluate the impact of peripheral injection of atomoxetine on the recovery time of midazolam, three trials were done for each group of mice (vehicle, 10 mg/kg, and 20 mg/kg), and the recovery time was recorded.

To further verify the effect of NE on the recovery time of midazolam, we used DSP-4 (C8417, Sigma-Aldrich), a selective neurotoxin that targets the LC NE system in the rodent brain and is used to disrupt nerve terminals and attenuate NE and NE transporter function in the LC-innervated brain regions. We divided the mice into four groups (vehicle + vehicle, vehicle + atomoxetine, DSP-4 [3 days] + atomoxetine, DSP-4 [10 days] + atomoxetine; n = 6 in each group). The specific experimental method followed was as follows. C57BL/6J mice received IP injection of DSP-4 (50 mg/kg) or saline (vehicle) 3 days before or 10 days before the IP injection of atomoxetine (20 mg/kg) or saline (vehicle). One hour later, midazolam was intraperitoneally injected, and the recovery time was recorded. Herein, we verified the physiological effects of DSP-4 using immunohistochemistry.

### Effects of intra-LC microinjection of DSP-4 and its interaction with IP injection of atomoxetine on the recovery time of midazolam

Ten days before the start of this part of the experiment, cannulas were implanted in bilateral LC of C57BL/6 J mice using the same method as mentioned above. DSP-4 (200 nL, 10 µg/µL) or saline (vehicle) was injected into the LC through these guide cannulas. After 10 days, the mice were intraperitoneally administered atomoxetine or saline (vehicle) 1 hr before the IP injection of midazolam. The recovery time was recorded for vehicle + vehicle, vehicle + atomoxetine, and DSP-4 + atomoxetine groups (n = 6 mice in each group). Using the same protocol as mentioned above, to complete the experiment, the mice were perfused and their brains were sectioned. Then, the number of TH+ cells

in the DSP-4-treated group and vehicle group was counted and compared individually to investigate the effects of DSP-4 microinjection on noradrenergic neurons in bilateral LC.

### Effects of lateral ventricle injection and intra-LC microinjection of a GABAA receptor antagonist on the recovery time of midazolam

This experiment was performed in the same batch of 8-week-old C57BL/6J mice 1 week after lateral ventricle cannula and bilateral LC cannula implantation. Three minutes before the IP injection of midazolam, 2000 nL of gabazine (2 µg/mL, 4 µg/mL, n = 7, HY-103533, MedChemExpress) was injected into the lateral ventricle cannula or the bilateral cannulas of LC. The recovery time was recorded in each group. During the experiment, the calcium signals were also recorded simultaneously.

### Effects of lateral ventricle injection of different adrenoceptor agonists and antagonists on the recovery time of midazolam

The experiment was performed in the same batch of 8-week-old C57BL/6J mice 1 week after lateral ventricle cannula implantation. The mice were administered agonists or antagonists of different adrenergic receptors through the lateral ventricle. Three minutes before IP injection of midazolam, 2000 nL of the α1 receptor agonist phenylephrine (10 mg/mL, 20 mg/mL, n = 8, HY-B0471, MedChemExpress) or α1 receptor antagonist prazosin (0.75 mg/mL, 1.5 mg/ml, n = 8, HY-B0193, MedChemExpress) or α2 receptor agonist clonidine (0.75 mg/mL, 1.5 mg/mL, n = 9, HY-B0409, MedChemExpress) or α2 receptor yohimbine (15 µmol/mL, 22.5 µmol/mL, 30 µmol/mL, n = 8, Y111137, Aladdin) or β receptor agonist isoprenaline (2 mg/mL, 4 mg/mL, n = 7, I5627, Sigma-Aldrich) or β receptor antagonist propranolol (2.5 mg/mL, 5 mg/mL, n = 6, HY-B0573, MedChemExpress) or vehicle was administered via the lateral ventricle catheter. The recovery time was recorded.

To further explore the interaction between the α1 receptor and β receptor, we divided the mice into four groups (vehicle + vehicle, phenylephrine, propranolol, and phenylephrine + propranolol). C57BL/6J mice received ICV injection of 2000 nL of phenylephrine (20 mg/kg) or propranolol (5 mg/kg) or phenylephrine (20 mg/kg) + propranolol (5 mg/kg) or vehicle, and 3 min later, they were treated with IP injection of midazolam. The recovery time was recorded.

### Effects of intra-LC microinjection of GABAA receptor antagonist and intra-VLPO microinjection of α1 receptor antagonist on the recovery time of midazolam

To investigate the interaction between GABAergic and noradrenergic systems, we divided the mice into four groups (vehicle + vehicle, gabazine + vehicle, vehicle + prazosin, and gabazine + prazosin). The experiment was performed 1 week after LC or VLPO cannula implantation. 200 nL of gabazine, prazosin, or vehicle in the same volume was microinjected into the LC or VLPO through the guide cannula. Three minutes later, the mice were intraperitoneally injected with midazolam. The recovery time was recorded.

## EEG recording and analysis

Mice with 2 weeks of viral infection in the target brain area were anesthetized with IP injection of 30 mg/kg pentobarbital and fixed on a stereotaxic apparatus to expose the skull for leveling. The top of the head was shaved for all mice, and then the skin was sterilized and dissected to expose the skull. Then, four screws with wires were drilled in the left and right of the fontanelle and in the left and right anterior of the posterior fontanelle, and these were then connected to the head stage where the EEG was recorded. This was followed by dental bone cement stabilization and the application of erythromycin ointment to the operative region. Mice equipped with an EEG head stage were housed individually for 1 week until they adapted to the recording cable and then moved to the experimental barrel, and the EEG activity of the mice was recorded using the EEG monitor and software.

## Immunohistochemistry

For immunohistochemistry analysis of the brain, the mice were euthanized after the experiments and their brains were carefully extracted from the skull. After perfusion with 4% paraformaldehyde in phosphate-buffered saline (PBS), the brains were saturated in 30% sucrose for 24 hr. Then, coronal

slices (thickness, 35 µm) were cut using a freezing microtome (CM30503, Leica Biosystems, Buffalo Grove, IL). Frozen slices were washed three times in PBS for 5 min and incubated in a blocking solution containing 10% normal donkey serum (017-000-121, Jackson ImmunoResearch, West Grove, PA), 1% bovine serum albumin (A2153, Sigma-Aldrich), and 0.3% Triton X-100 in PBS for 2 hr at room temperature. The sections were incubated with primary antibodies at 4°C overnight; this was followed by incubation in a solution of secondary antibodies for 2 hr at room temperature. The primary antibodies used were rabbit anti-TH (1:1000; AB152, Merck-Millipore), mouse anti-TH (1:1000; MAB318, Merck-Millipore), mouse anti-GABAA-R (1:500; ab94585, Abcam), and rabbit anti-c-Fos (1:1000, 2250T Rabbit mAb, Cell Signaling Technology, Danvers, MA), and the secondary antibodies used were donkey anti-rabbit Alexa 546 (1:1000; A10040, Thermo Fisher Scientific), donkey anti-mouse Alexa 546 (1:1000; A10036, Thermo Fisher Scientific), goat anti-rabbit Cy5 (1:1000; A10523, Thermo Fisher Scientific), donkey anti-rabbit Alexa 488 (1:1000; A21206, Thermo Fisher Scientific), and donkey anti-mouse Alexa 488 (1:1000; A21202, Thermo Fisher Scientific). The brain slices were washed three times with PBS for 15 min, and then they were deposited on glass slides and incubated in a DAPI solution at room temperature for 7 min. Finally, an anti-fluorescence attenuating tablet was applied to seal the slides. Confocal images were acquired using a Nikon A1 laser-scanning confocal microscope (Nikon, Tokyo, Japan), and further image processing was done using ImageJ (NIH, Baltimore, MD).

## Statistical analysis

All the experimental data were reported as mean ± SD. GraphPad Prism (GraphPad Software, Inc, San Diego, CA) and SPSS (SPSS Software Inc, Chicago, IL) were used for data visualization and statistical analysis, respectively. Before data analysis, all experimental data were subjected to the Shapiro–Wilk normality test. For comparative analyses of the two groups, if the data were normally distributed, Student's $t$-test, including independent samples $t$-test and paired samples $t$-test, was used. Conversely, if the data were non-normally distributed, the Mann–Whitney $U$ test or Wilcoxon signed-rank test was used. Notably, the Levene test was used to evaluate the homogeneity of variances. After the data met the normal distribution and homogeneity of variances, a one-way analysis of variance followed by Bonferroni's multiple comparison test was used for multiple comparisons. $p < 0.05$ indicates statistical significance (*Supplementary file 1*).

## Acknowledgements

The work was supported by the National Natural Science Foundation of China (grant no: 81771403, 81974205), the Natural Science Foundation of Zhejiang Province (LZ20H090001, LHZY24H090003), and the Program of New Century 131 outstanding young talent plan top-level of Hang Zhou to HHZ. We thank YuDong Zhou and Yi Shen for their help in experimental design.

## Additional information

### Funding

| Funder | Grant reference number | Author |
| --- | --- | --- |
| National Natural Science Foundation of China | 81771403 | HongHai Zhang |
| Natural Science Foundation of Zhejiang Province | LZ20H090001 | HongHai Zhang |
| Natural Science Foundation of Zhejiang Province | LHZY24H090003 | Yue Shen |
| National Natural Science Foundation of China | 81974205 | HongHai Zhang |

The funders had no role in study design, data collection and interpretation, or the decision to submit the work for publication.

## Author contributions
LeYuan Gu, Conceptualization, Investigation, Writing – review and editing; WeiHui Shao, Data curation, Investigation, Writing – original draft; Lu Liu, Qian Yu, Software, Investigation, Writing – original draft; Qing Xu, YuLing Wang, JiaXuan Gu, Yue Yang, XiTing Lian, Investigation; ZhuoYue Zhang, YaXuan Wu, Yue Shen, Investigation, Writing – original draft; HaiXiang Ma, Writing – original draft; YuanLi Zhang, Methodology; HongHai Zhang, Conceptualization, Resources, Data curation, Supervision, Funding acquisition, Validation, Project administration, Writing – review and editing

## Author ORCIDs
YaXuan Wu https://orcid.org/0009-0009-3484-0589
HongHai Zhang https://orcid.org/0000-0003-3530-2060

## Ethics
This study was approved by and all experimental procedures were performed in compliance with the Experimental Animal Ethics Committee of Zhejiang University (ZJU30701).

Reviewer #2 (Public Review): https://doi.org/10.7554/eLife.97954.3.sa1
Author response https://doi.org/10.7554/eLife.97954.3.sa2

---

# Additional files

## Supplementary files
• Supplementary file 1. Summary table of statistical analysis and reagents conducted throughout the study.
• MDAR checklist

## Data availability
Data are available at the Dryad website (https://doi.org/10.5061/dryad.2rbnzs7zs).

The following dataset was generated:

| Author(s) | Year | Dataset title | Dataset URL | Database and Identifier |
|---|---|---|---|---|
| Zhang H, Gu L, Hui S, Liu L, Xu Q, Wang Y, Gu J | 2024 | NE contribution to rebooting unconsciousness caused by midazolam | http://doi.org/10.5061/dryad.2rbnzs7zs | Dryad Digital Repository, 10.5061/dryad.2rbnzs7zs |

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
