## [Editor Report · eLife Assessment]

This study provides a **useful** set of experiments showing the relative contribution of the noradrenergic system in reversing the sedation induced by midazolam. The evidence supporting the claims of the authors is **solid**, although specificity issues in the pharmacology and neural-circuit investigations narrow down the strengths of the conclusions. Dealing with these limitations will make the paper attractive to medical biologists working on the neurobiology of anesthesia.

---

## [Referee Report · Reviewer #2 (Public Review)]

Summary:

This article mainly explores the neural circuit mechanism of recovery of consciousness after midazolam administration and proves that the LC-VLPO NEergic neural circuit helps to promote the recovery of midazolam, and this effect is mainly caused by α1 adrenergic receptors. (α1-R) mediated.

Strengths:

This article uses innovative methods such as optogenetics and fiber optic photometry in the experimental methods section to make the stimulation of neuronal cells more precise and the stimulation intensity more accurate in experimental research. In addition, fiber optic photometry adds confidence to the results of calcium detection in mouse neuronal cells.

This article explains the results from the entire system down to cells, and then cells gradually unfold to explain the entire mechanism. The entire explanation process is logical and orderly. At the same time, this article conducted a large number of rescue experiments, which greatly increased the credibility of the experimental conclusions.

Throughout the full text and all conclusions, this article has elucidated the neural circuit mechanism of recovery of consciousness after midazolam administration and successfully verified that the LC-VLPO NEergic neural circuit helps promote the recovery of midazolam.

The conclusions of this article are crucial to ameliorate the complications of its abuse. It will pinpoint relevant regions involved in midazolam response and provide a perspective to help elucidate the dynamic changes in neural circuits in the brain during altered consciousness and suggest a promising approach towards the goal of timely recovery from midazolam. New research avenues.

At the same time, this article also has important clinical translation significance. The application of clinical drug midazolam and animal experiments have certain guiding significance for subsequent related clinical research.

Comments on revised version: I have no further questions for this manuscript.

---

## [Author Response]

The following is the authors’ response to the original reviews.

**Reviewer 1：**
One major issue arises in Figure 4, the recording of VLPO Ca2+ activity. In Lines 211-215, they stated that they injected AAV2/9-DBH-GCaMP6m into the VLPO, while activating LC NE neurons. As they claimed in line 157, DBH is a specific promoter for NE neurons. This implies an attempt to label NE neurons in the VLPO, which is problematic because NE neurons are not present in the VLPO. This raises concerns about their viral infection strategy since Ca activity was observed in their photometry recording. This means that DBH promoter could randomly label some non-NE neurons. Is DBH promoter widely used? The authors should list references. Additionally, they should quantify the labeling efficiency of both DBH and TH-cre throughout the paper.

In Figure 5, we found that the VLPO received the noradrenergic projection from LC, indicating the recorded Ca2+ activity may come from the axon fibers corresponding to the projection. Similarly, Gunaydin et al. (2014) demonstrated that fiber photometry can be used to selectively record from neuronal projection.

We appreciate the reviewer's insightful suggestion to elaborate on the DBH promoter, we have now expanded our discussion to address the DBH (pg. 18): “DBH (Dopamine-beta-hydroxylase), located in the inner membrane of noradrenergic and adrenergic neurons, is an enzyme that catalyzes the conversion of dopamine to norepinephrine, and therefore plays an important role in noradrenergic neurotransmission. DBH is a marker of noradrenergic neurons. Zhou et al. (2020) clarified the probe specifically labeled noradrenergic neurons by immunolabeling for DBH. Recently, DBH promoter have been used in several studies (e.g., Han et al., 2024; Lian et al., 2023). The DBH-Cre mice are widely used to specifically labeled noradrenergic neurons (e.g., Li et al., 2023; Breton-Provencher et al., 2022; Liu et al., 2024). It is difficult to distinguish the role of NE or DA neurons when using the TH promoter in VLPO. Therefore, we used DBH promoter with more specific labeling. LC is the main noradrenergic nucleus of the central nervous system. In our study, we injected rAAV-DBH-GCaMP6m-WPRE (Figure 2 and 8) and rAAV-DBH-EGFP-S'miR-30a-shRNA GABAA receptor-3’-miR30a-WPRES (Figure 9) into the LC. The results showed that DBH promoter could specifically label noradrenergic neurons in the LC, while non-specific markers outside the LC were almost absent.”

As suggested, we have quantified the labeling efficiency of both DBH and TH-cre throughout the revised manuscript (Fig.2D; Fig.3D, N-O; Fig.4E-F, J, L; Fig.5E, L; Fig.6L, S, X; Fig.7G).

A similar issue arises with chemogenetic activation in Fig. 5 L-R, the authors used TH-cre and DIO-Gq virus to label VLPO neurons. Were they labelling VLPO NE or DA neurons for recording? The authors have to clarify this.

As previously addressed in response to Comment #1, we agree that it is difficult to distinguish the role of NE or DA neurons when using the TH promoter in the VLPO. Therefore, we injected the mixture of DBH-Cre-AAV and AAV-EF1a-DIO-hChR2(H134R)-eYFP/AAV-Ef1a-DIO-hM3Dq-mCherry viruses into bilateral LC and AAV-EF1a-DIO-hChR2(H134R)-eYFP/AAV-Ef1a-DIO-hM3Dq-mCherry virus into bilateral VLPO. Moreover, we quantified the labeling efficiency of DBH in the LC to demonstrate that this promoter can specifically label NE neurons (Fig. 5). Importantly, these corrections did not alter the outcomes of our results. Both photogenetic and chemogenetic activation of LC-NE terminals in the VLPO can effectively promote midazolam recovery (Fig. 5G, N).

Another related question pertains to the specificity of LC NE downstream neurons in the VLPO. For example, do they preferentially modulate GABAergic or glutamatergic neurons?

Our study primarily aimed to explore the role of the LC-VLPO NEergic neural circuit in modulating midazolam recovery. We acknowledge that our evidence for the role of LC NE downstream neurons in the VLPO, derived from activation of LC-NE terminals and pharmacological intervention in the VLPO (Fig.5, Fig.6, Fig.8, Fig.9) is limited. Accordingly, we now present the VLPO’s role as a promising direction for future research in the limitation section of our revised manuscript: “This study shows that the LC-VLPO NEergic neural circuit plays an important role in modulating midazolam recovery. However, the specificity of LC NE downstream neurons in the VLPO is not explained in this paper, which is our next research direction, VLPO neurons and their downstream regulatory mechanisms may be involved in other nervous systems except the NE nervous system, and the deeper and more complex mechanisms need to be further investigated.”

In Figure 1A-D, in the measurement of the dosage-dependent effect of Mida in LORR, were they only performed one batch of testing? If more than one batch of mice were used, error bar should be presented in 1B. Also, the rationale of testing TH expression levels after Mid is not clear. Is TH expression level change related to NE activation specifically? If so, they should cite references.

As recommended, we have supplemented error bar and modified the graph of LORR’s rate in the revised manuscript. (Fig. 1A-B; Fig. 9G-H).

We agree that the use of TH as a marker of NE activation is controversial, so in the revised manuscript, we directly determined central norepinephrine content to reflect the change of NE activity after midazolam administration (Fig. 1D).

Regarding the photometry recording of LC NE neurons during the entire process of midazolam injection in Fig. 2 and Fig. 4, it is unclear what time=0 stands for. If I understand correctly, the authors were comparing spontaneous activity during the four phases. Additionally, they only show traces lasting for 20s in Fig. 2F and Fig. 4L. How did the authors select data for analysis, and what criteria were used? The authors should also quantify the average Ca2+ activity and Ca2+ transient frequency during each stage instead of only quantifying Ca2+ peaks. In line 919, the legend for Figure 2D, they stated that it is the signal at the BLA; were they also recorded from the BLA?

In this study, we used optical fiber calcium signal recording, which is a fluorescence imaging based on changes in calcium. The fluorescence signal is usually divided into different segments according to the behavior, and the corresponding segments are orderly according to the specific behavior event as the time=0. The mean calcium fluorescence signal in the time window 1.5s or 1s before the event behavior is taken as the baseline fluorescence intensity (F0), and the difference between the fluorescence intensity of the occurrence of the behavior and the baseline fluorescence intensity is divided by the difference between the baseline fluorescence intensity and the offset value. That is, the value ΔF/F0 represents the change of calcium fluorescence intensity when the event occurs. The results of the analysis are commonly represented by two kinds of graphs, namely heat map and event-related peri-event plot (e.g., Cheng et al., 2022; Gan-Or et al., 2023; Wei et al., 2018). In Fig. 2, the time points for awake, midazolam injection, LORR and RORR in mice were respectively selected as time=0, while in Fig. 4, RORR in mice was selected as time=0. The selected traces lasting for 20s was based on the length of a complete Ca2+ signal. We have explained the Ca2+ recording experiment more specifically in the figure legends and methods sections of our revised manuscript.

To the BLA, we sincerely apologize for our carelessness, the signal we recorded were from the LC rather than the BLA. We have carefully checked and corrected similar problems in the revised manuscript.

**Reviewer 2：**
In figure legends, abbreviations in figure should be supplemented as much as possible. For example, "LORR" in Figure 1.

As suggested, we have supplemented abbreviations in figure as much as possible in the revised manuscript.

Additional recommendations:The main conceptual issue in the paper is the inflation of the conclusion regarding the mechanism of sedation induced by midazolam. The authors did not reveal the full mechanism of this but rather the relative contribution of NE system. Several conclusions in the text should be edited to take into account this starting from the title. I think the following examples are more appropriate: "NE contribution to rebooting unconsciousness caused by midazolam' or 'NE contribution to reverse the sedation induced by midazolam'.

As suggested, we have moderated the assertions about the mechanism of sedation induced by midazolam in several conclusions starting from the title (Line 1,125,150,169,202,237,482), to present a more measured interpretation in the manuscript.

Line 178-179, the authors state 'these suggest that intranuclear ... suppresses recovery from midazolam administration'. In fact, this intervention prolonged or postponed recovery from midazolam.

In our revised manuscript, we have corrected this inappropriate term (Line 178).

Pharmacology part (page 12) that aimed to pinpoint which NE receptor is implicated would suffer from specificity issues.In relation to the specificity issue, the focus on VLPO might be rational but again other areas are most likely involved given the pharmacological actions of midazolam.

In the revised manuscript, we have discussed those specificity issues of NE receptor and areas involved throughout the midazolam-induced altered consciousness: “In addition, given the pharmacological actions of midazolam, other areas may also be involved. Current studies suggest that the neural network involved in the recovery of consciousness consists of the prefrontal cortex, basal forebrain, brain stem, hypothalamus and thalamus. The role of these regions in midazolam recovery remains to be further investigated. Therefore, we will apply more specific experimental methods to determine the importance of LC-VLPO NEergic neural circuit and related NE receptors in the midazolam recovery, and conduct further studies on other relevant brain neural regions, hoping to more fully elucidate the mechanism of midazolam recovery in the future”.

Line 274, the authors used 'inhibitory EEG activity'. what does it mean? a description of which rhythm-related power density is affected would be more objective.Example of conclusion inflation: in line 477, the word 'contributes' is better than 'mediates' if the specificity issue is taken into account.

As suggested, we have improved our expression of words in our revised manuscript (pg. 13-14).

References

Gunaydin LA, Grosenick L, Finkelstein JC, et al. Natural neural projection dynamics underlying social behavior. Cell. 2014;157(7):1535-1551. doi:10.1016/j.cell.2014.05.017

Zhou N, Huo F, Yue Y, Yin C. Specific Fluorescent Probe Based on "Protect-Deprotect" To Visualize the Norepinephrine Signaling Pathway and Drug Intervention Tracers. J Am Chem Soc. 2020;142(41):17751-17755. doi:10.1021/jacs.0c08956

Han S, Jiang B, Ren J, et al. Impaired Lactate Release in Dorsal CA1 Astrocytes Contributed to Nociceptive Sensitization and Comorbid Memory Deficits in Rodents. Anesthesiology. 2024;140(3):538-557. doi:10.1097/ALN.0000000000004756

Lian X, Xu Q, Wang Y, et al. Noradrenergic pathway from the locus coeruleus to heart is implicated in modulating SUDEP. iScience. 2023;26(4):106284. Published 2023 Feb 27. doi:10.1016/j.isci.2023.106284

Li C, Sun T, Zhang Y, et al. A neural circuit for regulating a behavioral switch in response to prolonged uncontrollability in mice. Neuron. 2023;111(17):2727-2741.e7. doi:10.1016/j.neuron.2023.05.023

Breton-Provencher V, Drummond GT, Feng J, Li Y, Sur M. Spatiotemporal dynamics of noradrenaline during learned behaviour. Nature. 2022;606(7915):732-738. doi:10.1038/s41586-022-04782-2

Liu Q, Luo X, Liang Z, et al. Coordination between circadian neural circuit and intracellular molecular clock ensures rhythmic activation of adult neural stem cells. Proc Natl Acad Sci U S A. 2024;121(8):e2318030121. doi:10.1073/pnas.2318030121

Cheng J, Ma X, Li C, et al. Diet-induced inflammation in the anterior paraventricular thalamus induces compulsive sucrose-seeking. Nat Neurosci. 2022;25(8):1009-1013. doi:10.1038/s41593-022-01129-y

Gan-Or B, London M. Cortical circuits modulate mouse social vocalizations. Sci Adv. 2023;9(39):eade6992. doi:10.1126/sciadv.ade6992

Wei YC, Wang SR, Jiao ZL, et al. Medial preoptic area in mice is capable of mediating sexually dimorphic behaviors regardless of gender. Nat Commun. 2018;9(1):279. Published 2018 Jan 18. doi:10.1038/s41467-017-02648-0